# Development of a miRNA-Based Model for Lung Cancer Detection

**DOI:** 10.3390/cancers17060942

**Published:** 2025-03-10

**Authors:** Kai Chin Poh, Toh Ming Ren, Goh Liuh Ling, John S Y Goh, Sarrah Rose, Alexa Wong, Sanhita S. Mehta, Amelia Goh, Pei-Yu Chong, Sim Wey Cheng, Samuel Sherng Young Wang, Seyed Ehsan Saffari, Darren Wan-Teck Lim, Na-Yu Chia

**Affiliations:** 1Division of Respiratory Medicine, Sengkang General Hospital, Singapore 544886, Singapore; 2Molecular Diagnostic Laboratory, Tan Tock Seng Hospital, Singapore 308433, Singapore; 3Professional Officers Division, Singapore Institute of Technology, Singapore 828608, Singapore; 4Averywell Limited, Greater Manchester OL8 4QQ, UK; 5Duke-NUS Medical School, Singapore 169857, Singapore; 6National Cancer Center Singapore, Singapore 168583, Singapore

**Keywords:** lung cancer screening, biomarkers, microRNA, miRNA, low-dose computed tomography

## Abstract

Lung cancer remains the leading cause of cancer-related mortality worldwide, primarily due to late-stage diagnosis and limitations of current screening methods with low-dose computed tomography scans. Analysis of serum microRNA biomarkers may improve early detection of lung cancer. We aimed to discover a panel of miRNA biomarkers that is highly accurate in detecting lung cancer through a case-control study. A final panel of six miRNA biomarkers were selected with machine learning algorithms, achieving a high AUC of 0.86. The use of miRNA biomarkers shows promise in augmenting current lung cancer screening protocols and represent a significant step towards reducing lung cancer mortality.

## 1. Introduction

Lung cancer is the leading cause of cancer-related mortality worldwide, with approximately 1.8 million deaths annually [1]. Despite advancements in cancer therapeutics, including novel immunotherapies, lung cancer survival rates remain low [2,3]. This is primarily attributed to the asymptomatic nature of early-stage lung cancer, often resulting in late diagnosis [4]. Data from the Surveillance, Epidemiology, and End Results Program (SEER) indicates over half of lung cancer patients (53%) are diagnosed at a metastatic stage, where the 5-year survival rate is only 8.9% [5].

Early detection is critical for improving survival in lung cancer patients [6,7]. Low-dose computed tomography (LDCT) has proven to be a valuable tool for lung cancer screening, particularly among heavy smokers. The landmark National Lung Screening Trial (NLST) demonstrated a 20% relative risk reduction in lung cancer mortality with annual LDCT screening for heavy smokers aged 55–74 years [8]. This finding was also corroborated by another adequately powered lung cancer screening study, the NELSON trial [9].

Despite positive outcomes from these trials, widespread implementation of lung cancer screening with LDCT remains limited. LDCT screening is highly resource-intensive, requiring numerous medical specialists and readily accessible advanced imaging equipment [10,11,12,13]. Due to a high number needed to treat (NNT), the cost effectiveness of LDCT screening remains unclear. Additionally, LDCT has a high false-positive rate (7.9–49.3%) [14]. There are concerns regarding overdiagnoses, unnecessary investigations, risk from invasive biopsies and psychological distress amongst screened populations [15,16]. The limitations of LDCT highlight the need for adjunct tools to enhance lung cancer screening efficiency and accessibility.

Liquid biopsy, a minimally invasive diagnostic technique that analyses biomarkers in non-solid biological samples, has emerged as a promising approach for lung cancer detection. Various biomarkers, including circulating tumour cells (CTCs), circulating tumour DNA (ctDNA), cell free DNA (cfDNA), messenger RNA (mRNA) and microRNA (miRNA) have been evaluated for their association with lung cancer [17]. Among these, ctDNA and cfDNA assays are the most extensively investigated and are increasingly utilized for lung cancer treatment planning and post-treatment monitoring [18,19,20,21,22]. However, their sensitivity in early-stage cancer detection is limited due to the low abundance of circulating tumour DNA [21,22,23,24,25].

miRNAs demonstrate significant potential as biomarkers for the early detection of lung cancer. These small, non-coding RNA molecules (20–25 nucleotides) regulate gene expression at the post-transcriptional level by inducing translational repression or degradation of mRNA, thereby controlling key metabolic pathways and cellular processes [26,27,28]. The metabolomic alterations in lung cancer include disorders in glycolysis, the tricarboxylic acid cycle, fatty acid oxidation, glutaminolysis and amino acid synthesis [29,30,31]. The complex interplay between genetic alterations and metabolites derangement promotes tumorigenic processes such as cell proliferation, angiogenesis, migration, and resistance to apoptosis [32,33].

Unlike circulating DNA, miRNA dysregulation is frequently detectable in early-stage cancer, making them particularly valuable for early diagnosis [34,35]. Their stability in serum and ease of quantification using qPCR techniques further enhance their practicality for lung cancer screening [36]. miRNA sequencing analysis is usually less expensive as compared to ct-DNA and potentially a more cost-effective option for screening.

This study aims to develop and evaluate the predictive power of a serum miRNA-based model for lung cancer. By serving as an adjunct to current screening methods, the miRNA-based prediction model could significantly enhance early detection and improve lung cancer screening outcomes.

## 2. Materials and Methods

### 2.1. Study Design

We performed a case-control study in two tertiary teaching hospitals between 2020 and 2023. All participants were aged between 21 and 87 years of age and mentally competent to give informed consent for participation in the study. Patients with a known history of malignancy or active malignancy undergoing treatment were excluded from the study. Cases included patients with newly diagnosed lung cancer based on histological studies while controls included patients with no lung nodules, adenopathy, mass on radiographic imaging or benign lung nodules. A diagnosis of benign lung nodule was made following lung biopsy or if a lung nodule was less than 6 mm and had a less than 1% risk of malignancy.

Baseline patient characteristics and clinical features, including age, gender, ethnicity, smoking status, tumour site, stage, histology and molecular subtypes, lung nodule characteristics and radiological evidence of emphysema, were obtained from electronic health records.

### 2.2. miRNA Selection

Candidate miRNAs potentially involved in lung cancer pathogenesis were identified through a comprehensive literature review of major scientific databases, including PubMed, Embase, MEDLINE, Web of Science, Cochrane Library and ScienceDirect. The search employed key terms such as “miRNA”, “lung”, “pulmonary”, “cancer”, “malignancy”, “neoplasm” and “tumor”. To ensure a robust selection, we shortlisted miRNAs that had been reported in at least two independent studies and demonstrated functional relevance to tumorigenic processes like angiogenesis, apoptosis, invasion, and metastasis. The miRNAs expressed in a discovery cohort were subsequently tested in the final study cohort.

### 2.3. Sample Processing and miRNA Analysis

All subjects underwent venepuncture and had 5 mL of peripheral blood samples drawn in plain serum tubes (BD, Franklin Lakes, NJ, USA). Blood samples were left to clot for 45–60 min and centrifuged at 2000× *g* for 15 min at room temperature. Serum was aliquoted and stored in cryotubes at −80 °C for long term storage. If a blood sample was haemolyzed, the specimen was not analysed due to potential contamination by miRNAs released from red blood cells [37].

The expression levels of each miRNAs were quantified and reported as Ct values. Total RNA, including miRNA, was isolated from 200 µL of patient blood serum (Averywell, UK). Reverse transcription (RT) and qPCR were performed using Averywell’s mirLung Dx^TM^ Kit (Averywell, UK). TaqMan primers and probes were then used to quantitate each miRNA target via real time qPCR. The reactions were incubated in a 96-well plate at 95 °C for 30 s, followed by 40 cycles of 95 °C for 3 s and 60 °C for 30 s. The qPCR reaction was carried out in Applied Biosystems QuantStudio 3 and 5 (ThermoFisher Scientific, Waltham, MA, USA).

### 2.4. Statistical Analysis

Clinical characteristics between cases and controls were compared using two-sample *t*-tests for continuous variables and Chi-square tests for categorical variables, with statistical significance set at *p* < 0.05.

Clinical variables and miRNA expression levels were evaluated as predictive features and ranked separately using the Random Forest (RF) algorithm. The RF algorithm was chosen for its ability to handle complex interactions and nonlinear relationships, making it suitable for assessing the variable importance of miRNAs. RF’s ensemble structure enables a robust ranking of biomarkers based on their discriminative power while accounting for heterogeneous effects across variables. The clinical variables assessed included patient demographics (age, gender, race, smoking status and the presence of emphysema/COPD) and characteristics of lung nodules (number, size of the largest nodule, location and spiculation). The mean importance score from RF was used as a threshold, with miRNA biomarkers above this threshold selected for inclusion in the final risk model.

To evaluate the predictive performance of the top miRNA biomarkers and clinical variables, four machine learning models were employed: Neural Networks (NNET), Support Vector Machine (SVM), K-Nearest Neighbours (KNN), and Naive Bayes (NB). We selected KNN, NNET, SVM and NB because they represent a diverse set of machine learning approaches, balancing interpretability, flexibility and predictive performance. Each method has its own strengths and weaknesses: KNN is simple but sensitive to high-dimensional data, NNET captures complex patterns but requires careful tuning, SVM is robust to small datasets but computationally intensive, and NB is efficient but relies on strong independence assumptions.

The full dataset was used for training, and model performance was validated using five-fold cross-validation to mitigate overfitting. Hyperparameter tuning was also performed for each model through five-fold cross-validation to optimize performance.

Two feature sets were tested: (1) Model 1, consisting solely of the top biomarkers and (2) Model 2, which combined the top miRNAs with the size of the largest nodule. Model performance was primarily evaluated using the Area Under the Receiver Operating Characteristic Curve (AUC) as the accuracy metric. Additional performance metrics, such as sensitivity and specificity, were used to assess the model’s practical applicability.

Data analysis was performed in R software, R version 4.3.2 (R Core Team (2023). R: A language and environment for statistical computing. R Foundation for Statistical Computing, Vienna, Austria. URL https://www.R-project.org/ accessed on 31 October 2023.

## 3. Results

### 3.1. Patient Demographics and Clinical Characteristics

The study included 205 participants, with 82 lung cancer cases and 123 controls. The mean age of the participants was 60 years, with the majority being male (61%) and of Chinese ethnicity (75%). Lung cancer cases were older than the controls (66.5 vs. 57.0 years, *p* < 0.005) and were more likely to be heavy smokers (50 vs. 24.6%, *p* < 0.005). Emphysema was more commonly observed in lung cancer patients (15.9 vs. 7.3%, *p* = 0.016). This indicates age and smoking history as significant risk factors for lung cancer. Notably, malignant nodules were larger than benign nodules (40 vs. 15 mm, *p* < 0.005). Malignant nodules were predominantly located in the upper lobes of the lungs (48.8 vs. 19.5%, *p* < 0.005) and more commonly spiculated (31.7 vs. 2.4%, *p* < 0.005) (Table 1).

Adenocarcinoma was the predominant histological subtype (71%) of lung cancer. A significant proportion (39%) of lung cancer cases were never-smokers. In terms of cancer staging, most lung cancer cases were diagnosed at advanced stages (Stage 3: 9.8%, Stage 4: 48.8%). Small cell lung cancer (SCLC) was present in 8.5% of cases (n = 7) and were mostly diagnosed (71.4%) in the extensive stage (Table 1).

### 3.2. Identifying Top Clinical Features and miRNA Biomarkers for Lung Cancer Detection

In total, 25 candidate miRNAs were identified as potential lung cancer biomarkers after a comprehensive literature review. From this list, 16 miRNAs demonstrated a minimum two-fold increase in expression levels in a discovery cohort (n = 8) and were included in the full study cohort (Appendix A).

The RF analysis produced a ranked list of clinical features according to their predictive importance for lung cancer risk. Notably, the size of the largest lung nodule emerged as the most important clinical predictor, highlighting its strong association with malignancy. Additional significant clinical features included age, nodule spiculation, smoking status, race, presence of emphysema/COPD and gender, which were sequentially ranked in decreasing order of importance (Figure 1).

In parallel, RF analysis was also applied to the 16 miRNAs to evaluate their discriminative value. In total, 6 miRNAs—miR-196a-5p, miR-1268, miR-130b-5p, miR-1290, miR-106b-5p, and miR-1246 (Figure 2)—had RF importance scores above the mean index and were selected for further analysis. These six miRNAs displayed significantly higher expression in lung cancer cases compared to the controls, underscoring their potential role as key biomarkers in disease detection. The combined results of clinical features and miRNA rankings emphasize the importance of both nodule characteristics and specific miRNAs in predicting lung cancer risk.

### 3.3. Clinical Performance Index

Model 1 (miRNA-only): By using only the top six miRNA biomarkers, NNET achieved the highest AUC of 0.863, demonstrating strong discriminative ability for lung cancer based solely on miRNA expression (Figure 3). Machine learning using SVM achieved an AUC of 0.802, with KNN and NB showing slightly lower AUCs of 0.781 and 0.790, respectively (Appendix A). Sensitivity in Model 1 ranged from 70% to 78%, with NNET showing the highest sensitivity (0.775) and KNN the lowest (0.700). Specificity varied from 73% to 85%, with NNET again achieving the highest specificity (0.850) (Table 2).

Model 2 (miRNA + nodule size): Incorporating the largest nodule size as an additional feature significantly improved diagnostic performance. KNN achieved the highest AUC (0.989) in this configuration (Figure 4), followed closely by NB (AUC = 0.978). SVM and NNET also performed well, with AUCs of 0.983 and 0.961, respectively (Appendix A). Sensitivity in Model 2 ranged from 92% to 98%, with SVM and KNN attaining the highest sensitivity scores (0.975 and 0.921, respectively). Specificity increased to 93–98%, with KNN achieving the highest specificity (0.975), followed closely by NB (0.976). The substantial improvement in predictive performance in Model 2 underscores the added value of including clinical information, particularly nodule size, in a lung cancer risk prediction model (Table 2).

## 4. Discussion

### 4.1. Synergistic Role of a miRNA Panel in Lung Cancer Detection

The six miRNAs in the final panel enhanced diagnostic accuracy as they target interrelated oncogenic pathways implicated in the pathogenesis of lung cancer. miR-196a-5p and miR-106b-5p directly promote uncontrolled cell proliferation through the PI3K/AKT pathway [38,39,40]. Similarly, miR-130b-5p, miR-1246, and miR-1290 regulate epithelial-mesenchymal transition (EMT), a key process in metastasis, by altering expression of proteins such as DPP-4 [41] and metallothionein [42]. Additionally, miR-106b-5p and miR-1290 suppress pro-apoptotic genes such as BTG3 [43], allowing lung cancer cells to evade programmed cell death.

The six miRNAs also work synergistically through differential expression patterns across disease stages. miR-130b-5p, miR-1290 and miR-1246 are consistently overexpressed in advanced NSCLC and correlate with disease progression and cancer stemness [44,45]. miR-106b-5p and miR-130b-5p are elevated in early-stage lung cancer, making them suitable for early detection [44,46]. The integration of these miRNAs into a single diagnostic panel provides better specificity and sensitivity than any individual miRNA biomarker.

### 4.2. Limitations of Existing Lung Cancer Screening Strategy: Identifying Risk Amongst Non-Smokers

While LDCT screening reduces lung cancer mortality among heavy smokers, its utility for non-smokers and individuals with non-smoking-related risk factors remains unclear. This issue is especially relevant in Asia, where a substantial proportion (45–70%) of lung cancer cases occur in non-smokers [47,48,49]. A simulation study suggest that lung cancer screening limited to heavy smokers in Asian populations reduces mortality by only 3.76–4.74%, emphasizing the need to expand screening criteria to better capture high-risk populations [16].

Clinical prediction models offer a promising solution to improve lung cancer screening criteria. Tools such as PLCOm2012, the Liverpool Lung Project (LLP), the Lung Cancer Death Risk Assessment Tool (LCDRAT) and the Henan Lung Cancer Risk Model enable personalized risk assessments based on demographic and clinical factors. These models enhance lung cancer risk stratification, reducing the number needed to treat (NNT), and increasing the proportion of screen-preventable deaths [50,51,52,53,54]. However, these existing clinical prediction models are still heavily reliant on smoking history for risk assessment and may not accurately predict lung cancer risk in non-smokers.

Expanding LDCT screening criteria to include non-smokers with risk factors such as family history, passive smoke exposure, or chronic respiratory conditions has been shown to increase rates of lung cancer detection. The Taiwan Lung Cancer Screening in Never-Smoker Trial (TALENT) study demonstrated the effectiveness of LDCT in detecting lung cancer amongst non-smokers with these risk factors. The lung cancer incidence in the study was twice of that reported in the NLST and NELSON trials, underscoring the importance of incorporating individuals with non-smoking-related risk factors into lung cancer screening programmes [6].

### 4.3. Challenges in Management of Screen-Detected Lung Nodules

In addition, lung cancer screening with LDCT alone is associated with a high rate of false-positive results. Most screen-detected lung nodules are benign, yet they often necessitate further evaluation, including invasive biopsies, which contribute significantly to healthcare costs. A cost analysis study estimated that 43.1% of the total cost of lung cancer workup is attributed to invasive biopsies of benign lung nodules [55].

Initial risk assessment of lung nodules often involves the use of prediction models. Subsequent investigations, including functional imaging with positron emission tomography (PET) scans, are guided by the risk of malignancy (Appendix A). The Brock and Mayo Clinic prediction models are widely recommended in major guidelines, including American College of Chest Physicians [56], the British Thoracic Society [57], and Fleischner Society [58]. These models utilize demographic, clinical, and radiologic features, including patient age, gender, family history of lung cancer, nodule size and location, and the presence of emphysema to estimate risk of lung malignancy. While effective in estimating malignancy risk, these models have primarily been validated in heavy-smoking, predominantly Caucasian cohorts.

The predictive performance of these models in Asian or non-smoking populations remains uncertain [56,59]. Their reliability in assessing indeterminate nodules is moderate, with AUC values ranging from 0.67 to 0.70 [60]. PET scans, which are often used to refine malignancy risk in indeterminate nodules, also has many limitations. They are expensive, often not readily accessible, and have reduced diagnostic specificity in regions with high tuberculosis prevalence, including many parts of Asia [61].

### 4.4. Improving Lung Cancer Screening with miRNA-Based Prediction Models

The panel of serum miRNA biomarkers, with its ease of application and high predictive ability for lung cancer, could be utilised to identify high-risk individuals who may not meet conventional screening criteria (heavy smoking history). The miRNA test could identify individuals with non-smoking-related risk factors and improve their enrolment into lung cancer screening programs (Appendix A). Compared to traditional clinical models such as the PLCOm2012, the panel of six serum miRNA biomarkers exhibits superior discriminative power, which could translate into improved screening outcomes.

Integrating the serum miRNA biomarker panel with nodule size from LDCT also significantly enhanced predictive power, achieving an AUC of 0.989 (Figure 4). This approach is consistent with findings from the BioMILD trial, which demonstrated that combining LDCT with an miRNA signature classifier (MSC) outperformed conventional risk models, including LCDRAT, PLCOm2012, and the Brock model [62]. The combined model holds promise for optimizing lung nodule management, reducing the need for unnecessary invasive procedures, and improving the cost-effectiveness of lung cancer screening programs (Appendix A).

### 4.5. Study Strength and Limitations

This study addresses limitations of many prior research, including failure to include all lung cancer subtypes and using techniques such as RNA-Seq or microarrays that are less suitable for adoption in large-scale lung cancer screening [63,64,65]. We utilized a highly accurate qPCR method to quantify serum miRNA levels, included all lung cancer subtypes, and used machine learning to identify the optimal combination of miRNA biomarkers. These methodological choices enhance the reliability and applicability of our findings. The lung cancer cases in our study cohort closely reflected the regional epidemiology, characterized by a high proportion of non-smokers (40.2%) and a predominance of adenocarcinomas.

A study limitation was that majority of subjects were of South-East Asian ethnicity. This limits the generalizability of our findings, especially as lung cancer subtypes and genetic mutations vary across populations. Furthermore, we did not control for smoking status, a potential confounder in miRNA-based lung cancer detection [66]. The study included both early and late-stage cases due to limited sample size. While this demonstrates the test’s applicability across all stages of lung cancer, the sensitivity for detection of early-stage cancer needs to be affirmed in future studies with larger cohorts stratified by cancer stage.

### 4.6. Challenges and Future Directions for miRNA-Based Screening

While miRNA biomarkers show promise as a novel lung cancer screening tool, larger studies are required to validate technical reliability and diagnostic accuracy. With an improved understanding of lung cancer metabolomics, the interplay between miRNA and lung cancer metabolism represents a novel field for further research.

There is a vital need to close the knowledge gap between miRNA as potential lung cancer biomarkers and practical implementation in cancer-screening programs. Clear guidelines are needed for optimal management of false-positive miRNA results (e.g., cases with benign lung nodules), which may arise from pre-cancerous conditions, extrapulmonary cancers, or infections.

Another critical consideration is the cost-effectiveness of miRNA-based lung cancer screening. Although miRNA assays may improve diagnostic accuracy and reduce rates of false positives, they come with high costs. Comprehensive health economic analyses will be necessary to determine the feasibility of incorporating miRNA assays into lung cancer screening at a national or global scale.

## 5. Conclusions

Our study highlights the potential of using miRNA biomarkers to improve lung cancer screening, particularly when combined with LDCT. This approach may help address the limitations of current screening strategies and LDCT, offering a more efficient and effective means of detecting lung cancer. Further validation in larger, ethnically diverse cohorts is necessary before miRNA biomarkers can be integrated into routine clinical practice. Continued research will be critical to refining this approach and maximizing its impact on lung cancer detection.

## Figures and Tables

**Figure 1 cancers-17-00942-f001:**
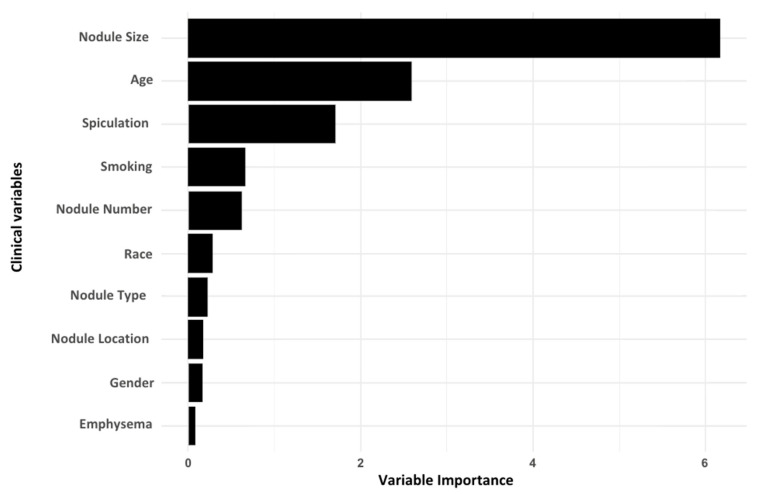
Clinical features selection with RF (training).

**Figure 2 cancers-17-00942-f002:**
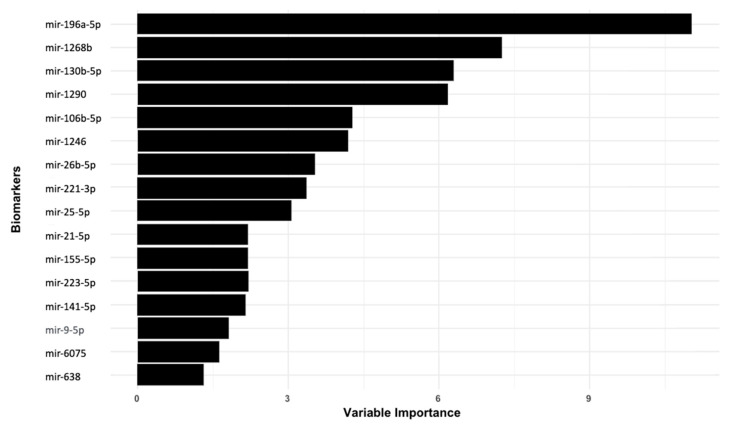
miRNA biomarkers ranking with RF (training).

**Figure 3 cancers-17-00942-f003:**
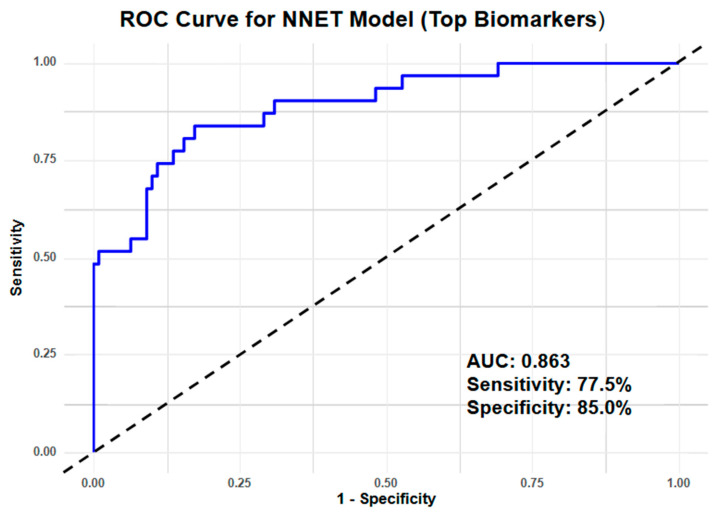
Machine learning using NNET showed the best-performing ROC curve for Model 1.

**Figure 4 cancers-17-00942-f004:**
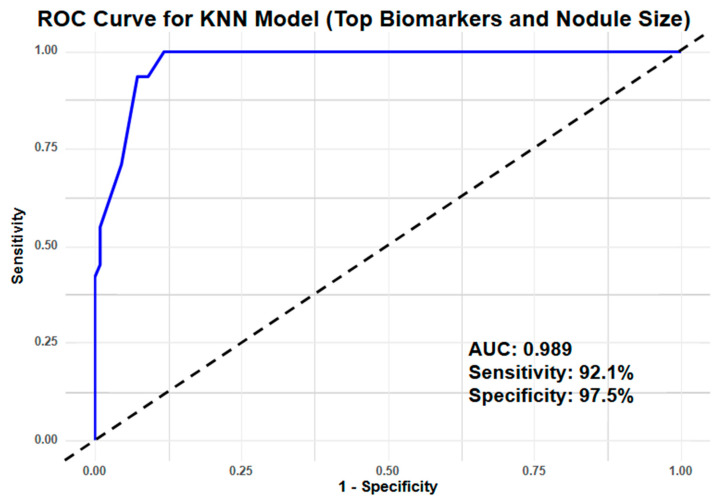
Machine learning using KNN showed the best performing ROC curve for Model 2.

**Table 1 cancers-17-00942-t001:** Patient demographics and clinical characteristics.

Clinical Characteristics	Controls (N = 123)	Cases (N = 82)	*p* Value
Patient demographics			
Age (mean ± SD)	57.0 ± 14.2	66.5 ± 10.7	<0.005
Gender (%)			
Male	72 (58.5)	54 (65.9)	0.17
Female	50 (40.7)	28 (34.1)	
Race (%)			
Chinese	95 (77.2)	59 (72.0)	0.23
Malay	14 (11.4)	10 (12.2)	
Indian	8 (6.5)	4 (4.9)	
Others	5 (4.1)	9 (11.0)	
Missing values			
Smoking history (%)			
Never smoker	89 (72.4)	32 (39.0)	
Smoker/ex-smoker	29 (23.6)	42 (51.2)	<0.005
Missing values	4 (3.3)	8 (9.8)	
Emphysema (%)			
Yes	9 (7.4)	13 (15.9)	0.018
No	110 (89.4)	61 (74.4)	
Missing values	4 (3.3)	8 (9.8)	
Cancer stage at diagnosis (%)			
1	N.A.	22 (26.8)	N.A.
2	N.A.	5 (6.1)	
3	N.A.	8 (9.8)	
4	N.A.	40 (48.8)	
Limited (for SCLC)	N.A.	2 (2.4)	
Extensive (for SCLC)	N.A.	5 (6.1)	
Nodule characteristics			
Number of nodules			<0.005
None	75 (61.0)	0 (0.0)	
Single	29 (23.6)	48 (58.5)	
Multiple (>1)	19 (15.4)	34 (41.5)	
Size (of the most suspicious/malignant nodule) in mm	14.7 ± 24.9	39.7 ± 27.6	<0.005
Nodule type (%)			
Ground glass opacity	5 (4.1)	4 (4.9)	<0.005
Solid	40 (32.5)	64 (78.0)	
Part solid	2 (1.6)	6 (7.3)	
No nodule	76 (61.8)	0 (0.0)	
Spiculation (%)			
Spiculated/lobulated	3 (2.4)	26 (31.7)	<0.005
Not spiculated	44 (35.8)	48 (58.5)	
No nodule	76 (61.8)	0 (0.0)	
Missing values	0 (0.0)	8 (9.8)	
Location (%)			
Upper lobe	24 (19.5)	40 (48.8)	<0.005
Non-upper lobe	23 (18.7)	34 (41.5)	
No nodule	76 (61.8)	0 (0.0)	
Missing values	0 (0.0)	8 (9.8)	
Histology (%)			
Adenocarcinoma	N.A.	57 (69.5)	N.A.
Squamous cell carcinoma	N.A.	11 (13.4)	
Small cell lung cancer	N.A.	7 (8.5)	
Other malignancies	N.A.	6 (7.3)	
Nodule biopsied, benign	9 (7.3)	N.A.	
Nodule not biopsied	38 (30.9)	N.A.	
No nodule	75 (61.0)	0 (0.0)	

**Table 2 cancers-17-00942-t002:** Clinical performance index of the risk models using ML algorithms.

ML Method		Model 1	Model 2
KNN	AUC	0.781	0.989
	Sensitivity	0.700	0.921
	Specificity	0.732	0.975
NNET	AUC	0.863	0.961
	Sensitivity	0.775	0.937
	Specificity	0.850	0.929
SVM	AUC	0.802	0.983
	Sensitivity	0.762	0.975
	Specificity	0.732	0.937
NB	AUC	0.790	0.978
	Sensitivity	0.712	0.912
	Specificity	0.787	0.976

## Data Availability

Dataset available on request from the authors. The raw data supporting the conclusions of this article will be made available by the authors on request.

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
