# Peer review of "Development of a miRNA-Based Model for Lung Cancer Detection"

_cancers, 2025, doi:10.3390/cancers17060942_

Round 1

Reviewer 1 Report

Comments and Suggestions for Authors

By building machine learning models combining LDCT and miRNA expression in lung cancer patients, authors studied the miRNA potential as biomarkers on diagnosis of lung cancer. A case-control study was conducted with less than 100 patients and over 100 controls. By combining top miRNAs and nodule sizes while building prediction models, authors found over 90% sensitivity and specificity. Comments are below.

  1. Reference for false-positive rate.

The saying of “LDCT has a high false-positive rate (7.9-47.3%)” were mentioned twice in the manuscript without any reference. Please add sources of the high false-positive rate of LDCT.

  1. Figure 1. Line 186. “The nodule location (upper vs. lower lobe) was the second most significant predictor” However, in figure 1, nodule location was not shown as the second significant.
  2. Results. Line 196-198.

“These six miRNAs displayed significantly higher expression in lung cancer cases compared to controls, underscoring their potential role as key biomarkers in disease detection”. How’s the miRNA expression levels? Why authors using RF important score to rank miRNAs instead of patient miRNA expression levels comparing with controls?

  1. Model 1 and 2.

When adding nodule information in model 2, scores were significantly improved. I have the feeling that this was because all patients had at least one nodule (from table 1). Especially authors use the largest nodule size, which has strong correlation to lung cancer.

Have authors tried to use nodule size in model 1 and add miRNA information in model 2?

  1. Different fonts were found in the manuscript.
  2. Discussion line 270-274.

Based on the false-positive issue of LDCT, did the model help lower the false positive rate?

Since the model built based on both liquid biopsy and LDCT, which also raise the healthcare cost.

Reviewer 2 Report

Comments and Suggestions for Authors

This is a  well crafted manuscript which nicely demonstrates the importance of careful planning and organization.  Results are intriguing and hypothesis generating.  Limitation of the small size of this study is acknowledged.  This is important  This is a well written manuscript but lacks tumour size-related information on the studied cases as mentioned. Explanations of the  inherent strengths and weaknesses in the different machine learning modules would be of interest.  Suggest you add information regarding role of metabolomics which has been published and presented from different groups.  Could you re-analyze just the cases with small size tumour (1-2 cm) as the major challenge is to differentiate malignancy in small lesions?  Some idea of the estimated practical costs of miRNA analysis vs other technologies would be helpful considering the potential use in wide-spread population screening.

Round 2

Reviewer 1 Report

Comments and Suggestions for Authors

Authors responded to comments and responses were accepted by the reviewer. After modification, the manuscript is more comprehensive and reader friendly.

MiRNA biomarkers are promising screening tools for lung cancer. Authors combined patient miRNA expression with low dose computed tomography data, modified prediction models with higher prediction accuracy.

Suggest acceptance in the current manuscript format.